# Feasibility, Interrater Reliability and Internal Consistency of the German Environmental Audit Tool (G-EAT)

**DOI:** 10.3390/ijerph19031050

**Published:** 2022-01-18

**Authors:** Anne Fahsold, Kathrin Schmüdderich, Hilde Verbeek, Bernhard Holle, Rebecca Palm

**Affiliations:** 1Deutsches Zentrum für Neurodegenerative Erkrankungen e.V. Standort Witten, 58453 Witten, Germany; kathrin.schmuedderich@dzne.de (K.S.); bernhard.holle@dzne.de (B.H.); 2Department of Nursing Science, Faculty of Health, Witten/Herdecke University, 58455 Witten, Germany; rebecca.palm@uni-wh.de; 3Department of Health Services Research, Maastricht University, 6229 GT Maastricht, The Netherlands; h.verbeek@maastrichtuniversity.nl

**Keywords:** dementia-specific environment, environmental design, assessment instrument, long-term care, dementia, reliability, feasibility

## Abstract

Dementia-specific environmental design has the potential to positively influence capabilities for daily living and quality of life in people with dementia living in nursing homes. To date, no reliable instrument exists for systematically assessing the adequacy of these built environments in Germany. This study aimed to test the adapted version of the Environmental Audit Tool—High Care (EAT-HC)—the German Environmental Audit Tool (G-EAT)—with regard to its feasibility, interrater reliability and internal consistency. The G-EAT was applied as a paper-pencil version in the German setting; intraclass correlation coefficients at the subscale level ranged from 0.662 (III) to 0.869 (IV), and 42% of the items showed at least substantial agreement (Cohen’s kappa ≥ 0.60). The results indicate the need to develop supplementary material in a manual that illustrates the meaning of the items and practical implications regarding dementia-specific environmental design. Furthermore, the intersectionality of built and physical environments must be considered when interpreting G-EAT results in future research and applications to residential long-term care practice.

## 1. Introduction

Although 80.2% of the 4,127,605 people in need of care in Germany receive support at home from relatives (64.0%) and outpatient services (29.7%) [1], residential long-term care is an irreplaceable pillar of the German care system. A total of 14,494 nursing homes provided 877,162 full-time long-term care places in 2019 [2]. Hoffmann et al. (2014) identified that 51.8% (95%-CI 50.4–53.3) [3] of nursing home residents are diagnosed with dementia. Beyond this, the number of residents living with mild cognitive impairments or undiagnosed dementia cannot be quantified, but it is estimated to be much higher. Residential long-term care facilities in Germany differ in various respects, including in terms of their size, the number of full-time long-term care places offered and their living concepts. For example, the majority of nursing homes provide care in integrative living units where people with and without dementia live together. Only approximately 30% of living units exclusively accommodate people with dementia and are designated dementia special care units [4]. Nursing homes in Germany are often organized in living units—most with 2–3 per facility—but their characteristics are not clearly defined and vary across facilities.

The built environment has been known for several years to be a key element of dementia-specific care [5]. If nursing homes are designed with people with dementia in mind, this may help residents maintain their independence in daily activities and improve their quality of life [5]. Specifically, the advantages of such an adapted environment have been addressed in various literature reviews in recent years [5,6,7,8]. Several factors, such as special layout and oriental cues or a homelike environment, can influence this support in a positive way. These environmental factors have been examined in a number of studies and have led to the development of various design guidelines and assessment instruments [9,10,11]. To describe dementia-specific environments in detail, systematic assessment tools are needed to map the different aspects of dementia-specific design. Thus far, no instrument of this kind exists in Germany. In reviewing available tested and established English language instruments, the Environmental Audit Tool—High Care (EAT-HC) [12] showed the most similarities to the target setting regarding understanding of care of people with dementia [13].

The EAT-HC is the enhanced version of Fleming and Bennett’s Environmental Audit Tool (EAT). In 2011, the creators developed the EAT based on a literature review [8,14]. The ten subscales of the instrument constitute the so-called “Key Design Principles”: (1) risk reduction, (2) human scale, (3) seen and be seen, (4 and 5) positive and negative stimulation levels, (6) movement and engagement, (7) familiarity, (8 and 9) variety of places in and around the unit, and (10) design and way of life [12] (pp. 198–200). The subscales contain between 2 and 19 items for which a subscore is calculated. All items, with the exception of those of Key Design Principle “Design and way of life”, form the unweighted overall score of the EAT-HC. The items “What is the vision/purpose of the unit for people with dementia?” and “How well does the built environment enable this to happen?” [12] (p. 264) serve to explain a health care team’s goals for the particular living unit to be assessed. The results of the instrument reflect these goals, and the need for action (e.g., rebuilding of outside areas or improvement of wayfinding markers) may be deduced accordingly. For the EAT-HC, another 56 items were formulated [14,15,16]. In testing interrater reliability, 50 items with an interrater agreement of <70% or Cohen’s kappa < 0.2 were removed, resulting in a total of 77 items for the EAT-HC [16]. The majority of the instrument items are dichotomous and require a “Yes” or “No” response (*n* = 27). For 13 items, selection based on a three- to four-point categorical response scale is possible. In addition, for a number of items, the option “Not applicable” (*n* = 32) can be selected or an additional point awarded under Key Design Principle “Risk reduction” if the design of an environmental safety feature is unobtrusively applied (*n* = 6). For each subscale score, individual scores are added and converted into percentage results [14]. The EAT-HC was tested on a range of psychometric performance criteria. In a systematic review of tools for assessing the physical environment in health care, Elf et al. (2017) rated structural validity as fair and cross-cultural validity as good based on the COSMIN checklist [11,17]. With regard to content validity, the creators of the EAT-HC refer to the development of items based on a systematic review [8] and the EAT [9]. Pearson correlation was calculated between the overall score of the EAT-HC, the Therapeutic Environment Screening Survey for Nursing Homes (0.72), and the Special Care Unit Environmental Quality Scale (0.34) to evaluate concurrent validity (*p* = 0.01) [16]. Interrater reliability was calculated using the assessments of two unexperienced raters from a convenience sample of 30 nursing homes. The interclass correlation coefficients (ICCs) for each subscale varied between 0.52 and 0.99. Internal consistency was assessed with Cronbach’s α values ranging from 0.57 to 0.88 and was interpreted as satisfactory [16]. The instrument was developed for application in research [18,19] and environmental consultation at Dementia Training Australia.

The EAT-HC was translated and culturally adapted for the German context as part of a project of the Deutsches Zentrum für Neurodegenerative Erkrankungen e.V. [13]. In accordance with the adapted versions of the EAT-HC for Singapore [20] and Japan [21], a multistep translation process of the instrument for the German context took place according to the guidelines of the World Health Organization (1998) [22]. Potential users of the future instrument from dementia-related health care research and dementia care were involved in the process. Additionally, an evaluation of content validity calculating the content validity index [23] twice took place as part of the adaptation phase. Content validity at the item level (I-CVI) varies between 0.33 and 1.0 regarding relevance in the German long-term care setting and between 0.38 and 1.0 in terms of comprehensiveness. At the subscale level, the scores range between 0.69 and 0.99, and 0.57 and 0.97, respectively. Three items of Key Design Principle “Risk reduction” are controversial and will only be applied in secure living units in Germany [13]. Consequently, two versions of the instrument—for secure living units with 77 items and for nonsecure living units with 74 items—are available for the test procedure described in this article. Although the original Australian instrument shows satisfactory psychometric quality, the EAT-HC’s performance needs to be rechecked after cross-cultural adaptation. Since no German gold standard for assessing the built environment exists, neither concurrent validity nor parallel test reliability can be determined. Consequently, this study focuses on interrater reliability. In addition, we assess acceptance of data collection, time-related aspects of the procedure and practicability. Polit and Beck (2017) summarize these aspects under the term “feasibility”—a smaller test before beginning with a larger research project [24]. This article follows the *Guidelines for Reporting Reliability and Agreement Studies* (GRRAS) [25].

## 2. Materials and Methods

### 2.1. Design

The study involved a feasibility and interrater reliability study. First, the feasibility study examined acceptance of data collection, the time-related aspect of the procedure and its practicability. Second, the reliability study involved testing interrater reliability at the item and subscale levels as well as internal consistency.

### 2.2. Sampling and Participants

The sample size for measuring interrater reliability was computed using the EAT-HC calculation according to the recommendations of Steiner and Norman (1995) and the methodological guidelines of Walter et al. (1998) [16,26,27]. As information on proportions used in the original Australian templates is not reported, models were calculated for different distributions of characteristics.

For the feasibility study, three nursing homes from the first author’s professional network in one urban area and two rural areas participated. For the reliability study, we used data from a convenience sample of living units: 170 nursing homes within a 20 km radius of the research institute received a request to participate in the study. Upon initial contact, we checked whether the facilities had a secure living unit, as we focused on testing all 77 items of the G-EAT. If the requested facilities had such a living unit, this unit was included in the study. If all units appeared to be open, the facility managers decided for a given living unit how the assessment could least disturb the residents’ daily routines.

### 2.3. Instruments

#### 2.3.1. German Environmental Audit Tool

The German Environmental Audit Tool—High Care (G-EAT) is the adapted version of the above characterized EAT-HC [16]. Deviating from the original, we apply two versions of the G-EAT in this study: a version for nonsecure living units that contains 74 items and a 77-item version for secure living units [13].

#### 2.3.2. G-EAT Context Questionnaire

This questionnaire contains different items regarding (a) structure, (b) architecture, (c) financing, (d) staff, (e) residents and (f) meals. Some items have been applied in the study DemenzMonitor [28], and others are newly developed and were tested within the feasibility study. We applied this questionnaire to describe the context of the participating living units, but it did not serve as part of the interrater reliability testing.

#### 2.3.3. Additional Measurements to Test Feasibility

To assess the feasibility of the G-EAT, we measured the time spent explaining, assessing and reflecting the assessment with staff members in nursing homes. We focused on the well-being of the residents upon assessing and entering their living units. Additionally, practicability in terms of the tool’s format was checked with a paper-pencil version versus a digital version on a tablet.

### 2.4. Data Collection

To test feasibility, the G-EAT was conducted by two researchers (AF and RP) and at least one staff member of the respective nursing home. Allied staff members were familiar with the living unit to assess the group of residents. First, the G-EAT and Key Design Principles were introduced, and then the staff member was asked to show the living unit, additional spaces used by the group of residents and the outdoor area. Afterward, all raters—from research and practice—created a common definition of the living unit to delineate the boundaries of the unit. Every rater filled in the G-EAT independently while walking around the living unit. The researchers took field notes when they identified difficulties in terms of disturbing residents or staff through data collection (e.g., during meals in common areas, group activities or conversations). After the assessment, questions on the G-EAT that could not be answered by the researchers themselves were discussed with the staff member (e.g., regarding residents’ rooms). In addition, the staff member was asked about his or her experiences regarding disrupting the daily routine of the residents. Results from the feasibility study were included through team discussion processes into the interrater reliability data collection.

To test interrater reliability, data collection took place from August to December 2019 at the participating nursing homes. Two raters (AF and KS) with a background in nursing science as well as training as registered nurses conducted the assessment. Both were familiar with the instrument from various training sessions. In addition, the creators of the EAT-HC had trained AF during the instrument adaptation process. To allow independence of assessment, raters did not talk to each other about their assessments during data collection and, if possible, started their assessments in different areas in the living unit.

To avoid interrupting the residents’ meals, assessments were conducted between 9:00 a.m. and 11:30 a.m., and 1:00 p.m. and 4:30 p.m. Moreover, one to three staff members of the facility completed the instrument to further evaluate its feasibility. The results of their perspectives are reported elsewhere. After obtaining informed consent and explaining the project data, the scientific raters illustrated data collection with the G-EAT. Subsequently, the staff members guided the researchers through the living units, and all raters—from research and practice—found a common definition of the rooms to be visited. Raters were told to assess the built environment with the G-EAT independently and not discuss the items during data collection. If items remained open or questions could not be answered due to ethical considerations, the answers from the staff members were used. For these items, we are not able to compute the ICCs but the proportions of agreement.

### 2.5. Statistical Analysis

Interrater reliability was computed at the item and subscale levels. At the item level, unweighted Cohen kappa was used to calculate rater agreement [29]. At the subscale level, intraclass correlation coefficients were applied. In accordance with the McGraw and Wong convention (1996) [30], we utilize a formula to calculate *“two-may mixed effects, absolute agreement and single rater/agreement”* (Koo and Mae, 2016, p. 157) [31]. For internal consistency, we computed Cronbach’s alpha. In addition, proportions of agreement for all items were calculated (see Appendix A). Analyses were carried out using SPSS Version 25 (New York, NY, USA) [32].

### 2.6. Ethical Approval

The Ethics Commission of the German Society for Nursing Science approved this study (proposal number: 18-005). A letter from the study team informed staff and residents about data collection. Researchers only assessed shared spaces in the living unit and did not enter resident rooms without their invitation. Allying facility managers and staff members were asked to break up times of data collection if they sensed that the research staff would disrupt the daily routine of the residents.

## 3. Results

### 3.1. Participating Living Units

Forty nursing homes in the federal state of North Rhine-Westphalia participated in the interrater reliability study. The context questionnaire was only completed by a subset of the living units (*n* = 33). Table 1 shows their main characteristics: most were based in communities with 100,000 or more inhabitants (81.3%); only one facility was based in a more rural region (less than 20,000 inhabitants). The nursing homes varied in their numbers of units (2–10) and places (42–250). Only one unit of every nursing home was included. A total of 66.7% of the living units had an integrative living concept where residents with and without dementia lived together. Some living units were divided into smaller living groups by organization (37.5%) and/or spatially. In these cases, raters decided if they agreed with this from a spatial layout perspective, so only two living units were assessed with two living groups each.

Regarding architectural history, living units showed different characteristics: most had never been rebuilt (27.3%), or rebuilding had occurred more than 10 years ago (27.3%). The year of construction varied from 1933 to 2019.

### 3.2. Feasibility

The question regarding the practicability of the G-EAT assessment revealed that a paper-pencil version rather than a digital version was preferable for the IRR test. Comments and remarks on individual items can be added more easily, and sketches of living units or particular architectural features can be drawn on paper.

Acceptance of the data collection may be divided between residents’ and staff’s perceptions. Observations of the researchers and responses of the staff did not indicate that the presence of the researchers in the living unit assessing with the G-EAT disturbed the residents. With regard to their own acceptance, staff reported that they considered the questionnaire to be very long and that some items were difficult to interpret.

The duration of the assessments varied between 30 and 90 min (time from entering the living unit to all questions being answered). The length of the assessment not only depended on the size of the living unit but also on the related common areas (outdoor area, cafeteria, chapel, etc.) and their locations in the building. The use of these spaces at the time of assessment also mattered. For example, we did not enter commonly used rooms while residents ate lunch. Based on this knowledge, a period was chosen for IRRs during which residents did not take meals to avoid interrupting meals.

### 3.3. Interrater Reliability

For interrater reliability at the subscale level, ICC could be calculated for 8 out of 10 key design principles (see Table 2). The first key design principle cannot be calculated because of its qualitative nature. Key Design Principle II cannot be analyzed due to the small number of items involved (*n* = 2), so conclusions about reliability can be drawn only on an item level. ICCs range from 0.662 (III) to 0.869 (IV). According to Koo et al. (2016), four subscales showed moderate reliability, and two showed good reliability [31]. At the item level, Cohen’s kappa could be calculated for 63 items (see Appendix A). Items of Key Design Principle *“Environmental design as part of the care philosophy”* were not included, as agreement is the intended outcome. In addition, nine items could not be calculated due to a statistical problem, as one or both raters always chose one answer possibility. According to Landis and Koch (1977), 6 items showed poor agreement (K < 0.00), 6 showed slight agreement (K 0.00–0.20), 12 showed fair agreement (0.20–0.40), 9 showed moderate agreement (K 0.40–0.60), 22 showed substantial agreement (0.60–0.80) and 8 showed almost perfect agreement (K > 0.8). [33].

### 3.4. Internal Consistency

Cronbach’s alpha could not be computed for Key Design Principle I (qualitative character) and Key Design Principle II (small number of items). For the other subscales, scores varied from 0.362 (VII) (poor reliability) to 0.688 (IV) moderate reliability) [31] (see Table 3).

### 3.5. Modifications According to Psychometric Properties

To modify the items, we used the interrater reliability values as well as previously collected data on content validity [13] and information about practicability from the field, which were obtained from field notes of the first and second authors. Nineteen items were revised, and the gender form was simplified across the items due to the German division between female, male and diverse notation. Modifications of the items are based on several team discussions; initially, the first author suggested alternative formulations according to the test results. These were discussed with the other authors in different group sessions until a consensus was achieved. For three items, we were not able to find reformulations that could address their poor psychometric properties, so they were removed. Additionally, some items showed poor reliability, as they covered more than one aspect. We divided these and further developed three new items addressing culturally specific needs in the field of dementia-specific care detected during data collection (see Table 4). For example, an item measuring the possibility for overnight stays from family members at nursing homes was raised several times and seems to be an important factor in helping connect families.

## 4. Discussion

Our study aimed to test the feasibility, interrater reliability and internal consistency of the German Environmental Audit Tool. We found that the instrument is feasible in a paper-pencil version and that additional information according to the boundaries of the living unit is necessary before assessment. At the subscale level, the G-EAT demonstrated moderate to good reliability, and items with poor kappa values were discussed and modified. Internal consistency varies between subscales but with overarching room for improvement in item categorization. The identified test-theoretical weaknesses in the reliability of the G-EAT require reflection on the content and the different cultural backgrounds of the original instrument and its adapted version.

## 4.1. Reflecting Cultural Differences in Germany and Australia

A number of poor reliability scores were derived from the questionable validity of the items in the German setting. Although the G-EAT received extensive cultural adaptation during translation—involving experts from the field and research [13]—the interrater reliability values and contextual data indicate that some questions require further semantic and linguistic adaptation. In addition, ceiling effects, especially for items of Key Design Principles *“Support movement and engagement”* and *“Reduce risks unobtrusively”*, suggest that the differentiation of questions or a Likert scale [34] with more precise answers would be useful for the German setting. For example, the question of whether acoustic stimuli are used outside could be answered with “yes” for almost all living units. However, variance between the sound of a forest and a large-scale bird fountain could not be mapped with this tool. These results are supported by research on other cross-cultural adaptations of the G-EAT. Brennan et al. (2021) report for the adapted Japanese version (EAT-HC-JV) that cultural design elements such as foot baths or a tea ceremony room cannot be depicted with the original list of items [21]. Similar findings are reported by Sun and Fleming (2021) for the Singaporean version. To address this challenge, the author included additional items for the areas of palliative care, technology and spirituality in the Singapore Environmental Assessment Tool [20].

## 4.2. Addressing the G-EAT Characteristics for Research Purposes

Other items seem to not be applicable in our study due to their composition. For example, some questions ask for two constructs (e.g., orientation and stimuli), while others contain two questions (e.g., view of outside from living and dining rooms). This contradicts the methods of questionnaire construction [35]. Before testing interrater reliability, we did not change any of these items, as we aimed to keep the German questionnaire as similar as possible to the original Australian version to ensure cross-cultural comparisons. However, as the G-EAT shall provide precise conclusions regarding the built environment for research and health care teams at nursing homes, there is a need for further improvement and thus also to move away from the original instrument. The need to do so is underpinned by the results of an instrument evaluation by Quirke et al. (2021). The authors compared the Environmental Audit Tool (EAT) (previous version of the EAT-HC) to two other instruments concerning their use for planning, detailing and managing the built environment. The authors found that the EAT primarily (60%) facilitates the planning of long-term care facilities and only indirectly (12%) post-occupancy—the aspect most applicable to the purposes we focus on [36]. We initiated the required development of the G-EAT by removing inapplicable items and reformulating other cultural-specific questions to help draw a more accurate and complete map of the built environment. We also anticipate increasing the internal consistency of the data. However, this should be verified by appropriate procedures such as exploratory factor or Rasch analysis in the future.

## 4.3. Enhancing Understanding of the Dementia-Specific Design

In our study, two raters who were familiar with the G-EAT construct and with the relevance of the dementia-specific environment assessed the built environment. Nonetheless, the data collection results show that additional information should be provided to enable a more prescient response and thus interpretation of the G-EAT in future applications in research and long-term care practice.

Apart from the new items already presented and those that had to be removed, we modified the G-EAT to facilitate the interpretation of the items. For instance, all items were checked with regard to their readability, and phrasing-nested sentences that are inherent to the German language were reformulated to improve the items’ meaning. Furthermore, two additional elements were added to each item: notes on data collection and a comment field. Under *“Notes on data collection,”* we formulated instructions on how to answer an item on where the rater should stand when assessing visual axes. For example, acoustic stimuli in indoor areas are not applied, which might prevent stimuli from overwhelming residents [37]. Background information was added to describe the meaning of the question in more detail. This necessity is also pointed out by Sun (2021) and Brennan and colleagues (2021), who emphasize the different literacy levels of staff of facilities using the instrument to collect data [38]. Even for other instruments that depict the built environment, additional information is provided in manuals to address the difficulties of this difficult-to-capture construct [39,40,41].

## 4.4. Built vs. Social Environment

In addition to the cross-cultural and construct-related challenges described above, the fluctuating reliability values may also reflect a conceptual problem. Some points of raters’ disagreement arose due to raters attributing certain environmental aspects more to the architecture of the facility (built environment) or to the concept of the living unit (social environment). For example, a radio in a living unit can be mentioned here: it can offer a positive acoustic stimulus if it plays music known to the residents. To initiate this stimulus, staff usually turn on the radio, which represents a kind of dementia-specific intervention or serves as the beginning of a music group activity. The fact that these intersections between built and social environments have impacts and potential for people with dementia at home has already been explored [42]. Sun (2020) also reported that raters attribute or exclude different aspects of the built environment when testing the S-EAT [43]. This evidence from the literature shows that intersectionality—which represents an insurmountable challenge for test theory—cannot be neglected for dementia-specific care. Regarding G-EAT application, this means that a rater may allocate an unfulfilled item to the social environment and thus not systematically record it even though it is implemented.

## 4.5. Limitations of the Study

The study outlined here presents a few limitations that need to be considered when interpreting the results. We did not achieve the calculated sample size for all items, as staff and time resources did not permit the completion of such a comprehensive survey; hence, the sample size was set at 42 living units to ensure data collection for practical research reasons. As the results of this study must be considered against the clinical background, we may apply Cicchetti’s (1999) methodological recommendation that *“one should simply […] pay attention to the type of subject being assessed [and] train the examiners”* (p. 570) [44] if the intended sample size cannot be achieved. In addition, after data collection, the raters reflected on situations experienced in the field to assess the ethical aspects of assessing spaces while residents were present. Interference with the subsequent assessments could not be excluded but will be analyzed in more detail elsewhere. Since the general conditions for building and managing nursing homes vary between federal states, only nursing homes in one federal state of Germany participated. The different conditions in other states might affect the results of the G-EAT differently than those in the current sample. We also could not systematically capture a relevant source of bias in our study: seasonal changes in the environment. We conducted our surveys in different seasons but not repeatedly in the same living units. To measure the change sensitivity and interrater reliability of the G-EAT, a repeated-measures design—for example, as used by Calkins et al. (2007) [45]—would be useful in future studies.

Nonetheless, the study has strengths that should also be emphasized. Our comprehensive data collection and refinement of items based on additional data beyond interrater reliability values highlight that poor reliability often resulted from a lack of validity. This allows for this problem to be addressed without removing numerous items, thereby running the risk of altering or no longer being able to represent the underlying construct.

## 5. Conclusions

From the testing described in this paper, the German Environmental Audit Tool now exists in a modified version and may be applied in health care research. For our purpose of testing the G-EAT for its subsequent scientific use, our raters stood in for the population of interest as individuals with prior knowledge of the instrument’s content and handling. The usability and quality of the results for long-term care practitioners must be further evaluated with respect to the relevance of environmental design for practice and regarding the inclusion of this group in data collection. Given this background, we also plan to test the validity of the four new items in upcoming studies and intend to work on improving the internal consistency of the instrument with the implementation of the G-EAT in German health care research. Furthermore, the intersection of built and social environments poses a challenge to accurate data collection that should be considered in the course of the further application of the G-EAT in future surveys.

## Figures and Tables

**Table 1 ijerph-19-01050-t001:** Structural characteristics of the included living units in stage 2.

Characteristics (*n* = 33)	Sample
	%/M (*N*/Range)
**Sponsorship**	
nonprofit	67 (22)
profit	33 (11)
**Size of the community ***	
<20,000 inhabitants	3 (1)
20,000–100,000 inhabitants	16 (5)
100,000–1,000,000 inhabitants	81 (26)
**Structure of the facility**	
Number of units	4 (2–10)
Number of places (full-time)	96 (42–250)
**Year of construction ***	
<1945	3 (1)
1945–1959	6 (2)
1960–1979	22 (7)
1980–1999	19 (6)
2000–2010	28 (9)
>2010	22 (7)
**Time of last rebuilding**	
No rebuilding	27 (9)
Over the last 2 years	15 (5)
3–5 years ago	9 (3)
6–10 years ago	15 (5)
More than 10 years ago	27 (9)
Unknown time period	6 (2)
**Characteristics of included living units**	
Integrative living concept	67 (22)
Division into living groups	36 (12)
Number of resident rooms	25 (10–45)

* missing value (*n* = 1).

**Table 2 ijerph-19-01050-t002:** Interclass correlation coefficients on German Environment Audit Tool (G-EAT) subscale.

No.	Key Design Principle *	ICC (CI_95%_)	*p* Value	_N_Items	Interpretation of ICC Value **
II	Provide a human scale	-	-	2	
III	Reduce risks unobtrusively	0.662 (0.452–0.803)	<0.001	17	Moderate reliability
IV	Allow people to see and be seen	0.869 (0.769–0.927)	<0.001	10	Good reliability
V + VI ^1^	Manage levels of stimulation	0.728 (0.539–0.846)	<0.001	26	Moderate reliability
VII	Support movement and engagement	0.730 (0.505–0.854)	<0.001	9	Moderate reliability
VIII	Create a familiar place	0.698 (0.504–0.825)	<0.001	4	Moderate reliability
IX + X ^2^	Links to the community	0.712 (0.516–0.835)	<0.001	9	Moderate reliability

* according to Fleming & Bennett (2015) [16]; ** according to Koo et al. (2016) [31]; ^1^ contains two KDPs: “Reduce negative stimulus” and “Enhance positive stimulus”; ^2^ contains two KDPs: “In the living unit” and “In the community”.

**Table 3 ijerph-19-01050-t003:** Internal consistency of the G-EAT.

No.	KGP	_N_Living Units	Cronbach’s α	_N_Items
II	Provide a human scale	41	-	2
III	Reduce risks unobtrusively	41	0.568	13
IV	Allow people to see and be seen	42	0.688	10
V + VI ^1^	Manage levels of stimulation	41	0.353	25
VII	Support movement and engagement	42	0.362	9
VIII	Create a familiar place	42	0.503	4
IX + X ^2^	Links to the community	42	0.521	9

^1^ contains two KDPs: “Reduce negative stimulus” and “Enhance positive stimulus”; ^2^ contains two KDPs: “In the living unit” and “In the community”.

**Table 4 ijerph-19-01050-t004:** Overview of main item modifications.

Key Design Principle	Removed Items
III	Are different corridors clearly recognizable so residents can identify where they are?
III	Is the bed placed or can it be placed so that from lying down, the toilet seat can be seen?
V	Does each room have a distinctive character and atmosphere?
	New items
III	Can the exit leading to the outdoor area be seen from the dining room?
III	Do lying residents have a view to the outside from the dining room?
VII	Is there a clearly defined path to the outdoor area that avoids dead ends and locked exits?
VII	Are there sunny areas along the path in the outdoor area?
VII	Is there a shaded seating area in the immediate surrounding of the facility?
IX	Is there a space for private conversations in the living unit?
X	Is there a room within the facility for families to stay overnight?

## Data Availability

The data presented in this study are available on request from the corresponding author. The data are not made publicly available, as this would contradict the data use agreement reached with the participating nursing homes.

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
