# Peer review of "Feasibility, Interrater Reliability and Internal Consistency of the German Environmental Audit Tool (G-EAT)"

_ijerph, 2022, doi:10.3390/ijerph19031050_

Round 1
Reviewer 1 Report
The following is the summary of the present manuscript: Dementia-specific environmental design has the potential to positively influence capabilities for daily living and quality of life in people with dementia living in nursing homes. This study aimed to test the adapted version of the Environmental Audit Tool – High Care (EAT-HC) – the German Environmental Audit Tool (G-EAT) – with regard to its feasibility, interrater reliability and internal consistency. The G-EAT was applied as a paper-pencil version in the German setting, intraclass correlation coefficients on the subscale level ranged from 0.662 (III) to 0.869 (IV), and 42% of the items showed at least substantial agreement (Cohens kappa ≥0.60). The results indicate the need to develop supplementary material in a manual that illustrates the meaning of the items and practical implications regarding dementia-specific environmental design. Furthermore, the intersectionality of built and physical environments must be considered when interpreting G-EAT results in future research and applications to residential long-term care practice. The article is interesting. There is no doubt that the article contributes a lot to its belonging field. I have several minor comments: First, the authors can consider to reduce the volume of the introductions or divide them into fewer paragraphs. A total of eight paragraphs are too bulky. Second, in line 133, the authors can explain more about what “feasibility” studies are. Third, in line 155, I believe the term “times” should be revised to “time”. Fourth, in line 177, was there any procedure taken to make sure the independence between both raters? Fifth, please clarify which statistics had been used for quantifying reliability: ICC (shown in the results) or kappa (in the method)?Author Response
Dear reviewer,
The author team would like to thank you for their time and efforts invested in your. We deeply acknowledge your constructive, mind- and respectful revisions. We discussed all of the comments and reflected thoroughly how to address them.
We hope that we succeeded in doing so.

Reviewer 2 Report
The authors do an excellent job presenting their work with the appropriate background.
Following are some changes/comments suggested:
Line 16: Cohen's Kappa. Apostrophe is missing.
Line 34: For example, a majority of ..... not the majority of
Line 74: Usual convention is spell out numbers until nineteen and then use numerals, except at the beginning of a sentence.
This is a suggestion that the authors may ignore or just say that it is required for the paper to make sense. There is a lot of explanation on the EAT-HC. Is there a way to make the section more succinct?
There are inconsistencies with citations. When stating author last names, sometimes a year is provided and other times the year is omitted.
Table 1 size of community: Please follow , separators for thousands, instead of . separators. This is found in other places too. Please check for , as a decimal point or . as a thousand separator throughout the document.
Table 1, Year of Construction: If it is before 1945, it should be < 1945; as if saying Year < 1945. The reverse for 2010.
Table 2, KDP II cannot have a ICC due to low n. But, how was reliability deemed to be Good? Please explain.
General Comments on Overall Paper:
The authors have done a good job identifying the deficiencies of the EAT-HC as adapted to the German setting, especially the content validity issues. It would add a lot of value, if there were additional results based on the modification suggested in Table 4. G-EAT was intended to be used or adapted widely by others. It would be hard to do that with a survey or measurement instrument that does not have internal consistency/validity. I hope some work is being done in this area. The authors may highlight this in their conclusions.
Author Response
Dear reviewer,
The author team would like to thank you for their time and efforts invested in your. We deeply acknowledge your constructive, mind- and respectful revisions. We discussed all of the comments and reflected thoroughly how to address them.
